# A Neonatal Murine Model for Caprine Enterovirus Infection and the Viral Tissue Tropism

**DOI:** 10.3390/v15020475

**Published:** 2023-02-08

**Authors:** Qun Zhang, Fan Zhang, Xiaoran Chang, Junying Hu, Zhiyuan Zhang, Xuyuan Cui, Xuebo Zheng, Xinping Wang

**Affiliations:** State Key Laboratory for Zoonotic Diseases, Key Laboratory of Zoonosis Research of Ministry of Education, College of Veterinary Medicine, Jilin University, Changchun 130012, China

**Keywords:** caprine enterovirus infection, CEV-JL14, EV-G, mouse model

## Abstract

As the first caprine enterovirus identified from goat herds characterized by severe diarrhea with a high morbidity and mortality rate, the underlying pathogenesis and tissue tropism for CEV-JL14 remains largely unknown. Here, we reported the establishment of a neonatal murine model for caprine enterovirus and the unveiling of the tissue tropism and underlying pathogenesis for CEV-JL14 enterovirus. Susceptible murine strains, the infective dose, the infective routes, viral loads, and tissue tropism for CEV-JL14 infection were determined. The findings showed that ICR mice were susceptible to CEV-JL14 infection via all infection routes. Tissue viral load analysis showed that CEV-JL14 was detected in almost all tissues including the heart, liver, spleen, lung, kidney, intestine, brain, and muscle, with significantly higher viral loads in the heart, liver, lung, kidney, and intestine. These results revealed the pattern of viral load and tropism for CEV-JL14 and provided a model system for elucidating the pathogenesis of CEV-JL14 viruses.

## 1. Introduction

Enterovirus is a non-enveloped, positive single-stranded RNA virus that belongs to the genus *Enterovirus* within the family of *Picornaviridae* [1]. According to the latest virus classification by the International Committee on Taxonomy of Viruses, the genus of Enterovirus is composed of 12 species of enterovirus and three species of rhinovirus [1]. These viruses are pathogens that are related to neurological, respiratory, and digestive diseases in humans and animals [2]. Out of 12 enterovirus species, *Enterovirus* A (EV-A), *Enterovirus* B (EV-B), *Enterovirus* C (EV-C), and *Enterovirus* D (EV-D) are the major causative agents of human diseases such as hand-foot-mouth disease (HFMD), poliomyelitis, and myocarditis [3]. *Enterovirus* E (EV-E) and *Enterovirus* F (EV-F) are the causative agents of bovine enterovirus infections, characterized by digestive, respiratory, and neurological disorders [4,5]. The species *Enterovirus* G (EV-G) (formerly named *Porcine enterovirus B*) consists of twenty serotypes: EV-G1 to EV-G20, which are associated with infection in pigs and goats/sheep [6,7,8]. *Enterovirus* H (EV-H) and *Enterovirus* J (EV-J) primarily infects non-human primates [9,10]. In addition, *Enterovirus* I (EV-I), *Enterovirus* K (EV-K), and *Enterovirus* L (EV-L) are novel enterovirus species [1]. Although infections of enterovirus have been increasingly reported in cattle and pigs recently, enterovirus infections in small ruminants such as goats remain largely unknown. Previously, we reported the isolation of caprine enterovirus-JL14 (CEV-JL14), classified as EV-G, a novel caprine enterovirus from goats characterized by severe diarrhea with a high morbidity and mortality; developed a double sandwich ELISA for CEV detection; and uncovered the CEV infection in goat farms from different provinces in China [11,12,13]. However, the mechanisms underlying CEV infection remains largely unknown. Here, we report the establishment of a murine model for CEV infection, reveal the viral loads and tissue tropism, and uncover the histopathological lesions as such interstitial edema, necrosis, and lymphocyte infiltration in the heart, liver, spleen, lung, kidney, intestine, brain, and muscle in the mice infected by CEV-JL14, which will facilitate the studies on viral pathogenesis and immunity triggered by caprine enterovirus infection.

## 2. Materials and Methods

### 2.1. Ethics Statement

The experimental procedures for mice used in this study followed a standard protocol reviewed and approved by the Institutional Animal Care and Use Committee (IACUC) of Jilin University (approval no JLU-20150226), following strict compliance with the requirements of the Animal Ethics Procedures and Guidelines of the People’s Republic of China.

### 2.2. Mice and the Mouse Strain

Pregnant mice of BALB/c, ICR, and Kunming strains were obtained from Changchun Biological Products Institute. The mice were maintained in the laboratory animal facility of Jilin Province. All mice had free access to food and water and were kept in a temperature-controlled room (22 ± 0.5 °C) on reverse 12/12 h light/dark cycle. The new-born pups aged three days were randomly assigned to different treatment groups with each cage litter containing a dam and 5–10 pups.

### 2.3. Cell Culture and Virus Isolation

African green monkey kidney (Vero) cells were cultured in Dulbecco’s modified Eagle’s medium (DMEM) (Invitrogen, Carlsbad, CA, USA) containing 10% fetal bovine serum (HyClone, Beijing, China), 2 µg/mL gentamycin, and 2 mM L-glutamine in 5% CO_2_ at 37 °C. After being inoculated with CEV-JL14, Vero cells were cultured in DMEM containing 2% FBS, 2 µg/mL gentamycin, and 2 mM L-glutamine in 5% CO_2_ at 37 °C.

Virus isolation was performed as previously described [5]. Tissue samples from infected mice were homogenized in 10 mM phosphate-buffered saline (PBS) with a dilution of 1:10 (*w/v*) and centrifuged at 10,000× *g* at 4 °C for 10 min. After centrifugation, the supernatant passed through a 0.45 nm filter before inoculating the cells.

### 2.4. CEV-JL14 Virus and Infection of Neonatal Mice

CEV-JL14 virus was cultured and harvested in Vero cells as described previously [14]. Briefly, Vero cells with 70–80% of confluence were infected with 10^3^ TCID_50_ CEV-JL14, harvested at 48 h post infection, and kept at −80 °C. The stock virus was used for administration after titration.

Infection dose and routes for CEV-JL14 were performed as described previously [5]. For intramuscular injection and subcutaneous injection, 50 μL of virus suspension was injected into each mouse at two sites using a 31G ultra-fine hub-less insulin syringe (Beckton, Dickinson and Company, Franklin Lakes, NJ, USA). For intraperitoneal injection, 50 μL of virus suspension was administered. For intranasal and oral administrations, 50 μL of virus suspension was delivered through an STD Mouse Jugular Vein Cath system (Access™ technologies, Skokie, IL, USA). Mice were acclimatized for 10 min prior to handling, inoculation, or observation.

### 2.5. Necropsy and Tissue Collection

The mice used for this study were euthanized using cervical dislocation after CO_2_ inhalation. The euthanized mice were necropsied inside a Class II biosafety cabinet following the standard protocols. Tissue specimens collected for histopathological examinations were fixed in 4% formalin for 48 h at room temperature.

### 2.6. RNA Extraction, cDNA Synthesis, and PCR Amplification

Total RNA was extracted from Vero cells or tissue samples infected by CEV-JL14 with TRNzol kit (Tiangen, Beijing, China) in accordance with the manufacturer’s instruction. cDNA synthesis was performed using SuperScript^TM^Ⅱ Reverse Transcriptase (Invitrogen, Carlsbad, CA, USA) following the manufacturer’s instruction. PCR was performed using Taq DNA polymerase (Takara, Dalian, China) as described previously [15]. The primer sequences designed according to 5′UTR were listed as follows. CEV-JL14-UP: 5′-TGAACACAAACCGACCAATAG-3′; CEV-JL14-DN: 5′-TAATAAACAAATAAAGGAAACACG-3′.

### 2.7. Quantitation of Virus Loads in Mice Infected with CEV-JL14 Virus

Total RNA extraction and RT-PCR method specific for detecting CEV with a minimal detection of 1.13 × 10^3^ copies were performed as previously described [16]. Virus loads in different tissues of infected mice were quantitated with a real-time PCR method using Hieff^®^ qPCR SYBR Green Master Mix Kit following the manufacturer’s instructions (YEASEN, Shanghai, China).

### 2.8. Immunoperoxidase Monolayer Assay (IPMA)

IPMA was performed as previously described [15]. Vero cells were seeded in 24-well plates and were infected with the recovered CEV-JL14 viruses from infected mice. The uninfected Vero cells were used as the negative controls. At 24–48 h after infection, cells were fixed with ice-cold methanol at −20 °C for 30 min. Subsequently, cells were blocked in 5% skimmed milk and incubated with polyclonal antibody against CEV-JL14 (1:200 dilution) for 1 h at 37 °C [12]. After washing, the cells were incubated with horseradish peroxidase (HRP)-conjugated goat anti-rabbit IgG (1:500, Sigma, St. Louis, MO, USA) for 45 min at 37 °C. The plates were then stained with 3-amino-9-ethyl-carbazole (Amresco, Olympia, WA, USA) and captured by Canon digital camera (Canon, Tokyo, Japan).

### 2.9. Tissue Processing for Histopathological Analysis

Tissue samples for pathohistological analyses were processed following standard procedure as previously reported [5]. Samples fixed in formalin were dehydrated in an increasing ethanol series (50%, 70%, 80%, 95%, and 100% ethanol) at room temperature and twice with xylene before they were embedded in paraffin. The embedded samples were left to cool for 1–2 h, and a microtome was used to section the samples at 5 μm thickness.

### 2.10. Hematoxylin-Eosin (H&E) Staining and Immunohistochemistry Assay

H&E staining and immunohistochemistry assay were performed as previously described [5]. Briefly, H&E staining was carried out as follows: tissue sections were dewaxed by incubation in xylene, rehydrated in decreasing ethanol concentrations (100%, 95%, 70%, and 50%), and stained with hematoxylin and eosin to reveal the histopathological changes.

Immunohistochemistry assay was performed to detect the virus antigen in the tissues. After being dewaxed and hydrated, the slide was boiled for antigen retrieval in citrate buffer (pH 6.0) for 15 min, treated with 3% H_2_O_2_ for 15 min at room temperature, blocked in 5% skimmed milk, and incubated with polyclonal antibody against CEV-JL14 (1:200 dilution) for 1h at 37 °C, followed by staining with Rhodamine-conjugated goat anti-rabbit IgG (1:1000 dilution) for 45 min at 37 °C. The visualization signal was developed with 3-amino-9-ethyl-carbazole (AEC) and captured by CCD camera mounted on a Nikon epifluorescence microscope (Nikon Instruments Co., Ltd., Shanghai, China).

## 3. Results

### 3.1. ICR Mouse Strain Is Susceptible to CEV-JL14 Infection

To determine the infectivity of CEV-JL14 virus to mice, six neonatal mice from each of the ICR, BALB/c, and Kunming strains were randomly selected and used for CEV-JL14 virus infection, respectively. Each mouse was inoculated with 2 × 10^8^ TCID_50_ of CEV-JL14 virus by subcutaneous injection. The mice inoculated with the equal volume of DMEM were used as control. Five days after CEV-JL14 infection, mice in each group were euthanized, and tissue samples were collected for RNA extraction. CEV-JL14-5′UTR gene was amplified using RT-PCR. As illustrated in Figure 1, the fragment was amplified in ICR mice with an expected size. In contrast, no gene fragments were detected in BALB/c mice and Kunming mice (Figure 1). These results demonstrate that the ICR suckling mice strain is susceptible to CEV-JL14 virus infection. Therefore, ICR mice were selected for later study.

### 3.2. CEV-JL14 Virus Infects the ICR Suckling Mice via Various Infective Routes

To determine the infective routes of CEV-JL14, ICR suckling mice were inoculated with 50 μL of virus suspension containing 2 × 10^8^ TCID_50_ of CEV-JL14 via intraperitoneal injection, intramuscular injection, subcutaneous injection, oral administration, and intranasal administration. Five days after CEV-JL14 inoculation, the mice were euthanized and mixed tissue samples were collected for RNA extraction. The gene sequence of CEV-JL14 in the suckling mouse tissue was amplified using RT-PCR. As shown in Figure 2, fragments were detected in suckling mice inoculated with CEV-JL14 via all infective routes. These results demonstrate that suckling mice were infected with CEV-JL14 via intraperitoneal injection, intramuscular injection, subcutaneous injection, oral administration, and intranasal administration. Subcutaneous injection was selected for follow-up experiments.

### 3.3. The Infective Dose for CEV-JL14 to ICR Suckling Mice

To determine the minimal infective dose for CEV-JL14 to ICR suckling mice, ICR suckling mice were inoculated with different infective dose (2 × 10^4^, 2 × 10^6^, and 2 × 10^8^ TCID_50_) of CEV-JL14 by subcutaneous injection. Six mice were inoculated per dose. Mice in the control groups were inoculated with DMEM. Tissues including heart, liver, spleen, lung, kidney, intestine, and brain were harvested for RNA extraction after 5 days post-infection (dpi). The virus fragments were detected by RT-PCR. As shown in Figure 3, no fragment was detected in mice infected by 2 × 10^4^ TCID_50_ viruses, while fragments with expected sizes were detected in mice infected at doses of 2 × 10^6^ and 2 × 10^8^ TCID_50_. These findings suggested that the lowest infectious dose for three-day-old ICR suckling mice was 2 × 10^6^ TCID_50_.

### 3.4. Recovery and Characterization of CEV-JL14 Virus from the Infected Mice

To make sure that CEV-JL14 virus was indeed able to infect the mice, mixtures of tissue samples from infected ICR suckling mice were processed for virus isolation. As shown in Figure 4, Vero cells showed a typical cytopathic effect (CPE) in the 36–48 h after inoculation with tissue samples from infected mice (Figure 4A), while no CPEs were observed in Vero cells inoculated with normal mice tissue samples (Figure 4B).

To further confirm the CPEs observed in the above experiments, IPMA was performed on Vero cells with CPEs. Strong signals were observed in Vero cells infected with CEV-JL14 (Figure 4C), while no signals were observed in the control groups (Figure 4D). Further, quantitation of the virus loads in different samples for the infected mice using qPCR was shown (Figure 4E). Taken together, the above results further demonstrate that mice were indeed infected by CEV-JL14 viruses.

### 3.5. Persistence of CEV-JL14 in ICR Suckling Mice

To further ascertain the duration of CEV-JL14 in suckling mice, ICR suckling mice were injected with 2 × 10^6^ TCID_50_ CEV-JL14 by subcutaneous injection. Suckling mice in the control groups were injected with DMEM. Tissues samples were collected at 5, 11, 14, 16, 18, and 21 dpi, respectively. RT-PCR was performed to amplify CEV-JL14 genomic fragments. As illustrated in Figure 5, CEV-JL14 was detected persistently in infected ICR mice until 16 dpi. These results indicated that CEV-JL14 was present in ICR mice for at least 16 days.

### 3.6. Pathogenicity of CEV-JL14 to Suckling Mice

To explore the pathogenicity, multiple tissue samples from CEV-infected and uninfected ICR mice via different infection routes were collected and processed for H&E staining. As shown in Figure 6B, myocardial interstitial edema and lymphocyte infiltration was observed in the heart sections. A large number of inflammatory cells, mainly lymphocytes, were infiltrated in the liver, with some liver cells being necrotic and lysed (Figure 6D). Increased lymphocytes and necrotic cells were observed in the spleen with the inflammatory cell infiltration in red pulp (Figure 6F). Alveolar walls were significantly thickened by the inflammation with obvious pathological changes found, such as microvascular congestion and hemorrhage in the lung (Figure 6H). Severe lymphocyte infiltration was observed in the kidney, with granular degeneration in the renal tubular epithelium (Figure 6J). Intestinal villus interstitial edema and broken intestinal villus were revealed in the small intestine (Figure 6L). In addition, vacuolar degeneration was observed in the epithelial cells of the intestinal villi (Figure 6L). Vacuolar degeneration and necrosis were also observed in neurons (Figure 6N). Mild interstitial edema and extensive lymphocyte infiltration were found in muscle (Figure 6P). The severity of the pathological lesions observed in mice infected with CEV-JL14 via different infection routes are listed in Table 1.

### 3.7. Tissue Tropism for CEV-JL14

To investigate the tissue tropism of CEV-JL14, multiple tissue samples were collected from CEV-infected and uninfected ICR mice and processed for immunohistochemistry assay. As displayed in Figure 7, CEV-JL14 antigens were detected in the majority of the tissues, including the heart (Figure 7B), liver (Figure 7D), spleen (Figure 7F), lung (Figure 7H), kidney (Figure 7J), intestine (Figure 7L), brain (Figure 7N), and muscle (Figure 7P). It is interesting to note that CEV-JL14 antigens were abundantly detected in cardiomyocytes in the heart (Figure 7B), hepatocytes in the liver (Figure 7D), alveolar epithelial cells in the lung (Figure 7H), and the epithelial cells of distal convoluted tubules in the kidney (Figure 7J). Additionally, CEV-JL14 antigens were found in the red pulp of the spleen (Figure 7F) and distributed diffusely in the intestinal villi and epithelial cells of the intestinal mucosa (Figure 7L) and the muscles (Figure 7P). Furthermore, CEV-JL14 antigens were detected in a small number of cells in the ventricle and glial cells of the brain (Figure 7N). The distributions and strengths of CEV-JL14 antigen in mice infected via various routes are evaluated and summarized in Table 2.

Taken together, the above results demonstrate the distribution and reveal the difference of tissue tropism for CEV-JL14 strains.

## 4. Discussion

Caprine enterovirus infection is an emerging infectious disease that severely impacts the development of the goat industry [11]. Currently, the mechanism underlying CEV infection remains largely unknown due to the paucity of the animal model system. In this study, we established a murine model system and determined the susceptible mouse strain, the minimal infective dosage, the infection routes for the CEV-JL14 virus strain, and revealed the tissue tropism for CEV-JL14 viruses.

Mouse model systems were widely used in the study on the virus pathogenicity and viral pathogenesis [17,18,19,20]. In attempt to explore the feasibility of the mouse model for caprine enterovirus infection, we utilized CEV-JL14 virus to inoculate the neonatal mice of different strains, and found that the ICR mouse strain was susceptible to CEV-JL14 with a minimal dosage of 10^6^ TCID_50_. Moreover, we demonstrated that ICR neonatal mice were infected by CEV-JL14 enterovirus via all infection routes as such intraperitoneal injection, intramuscular injection, subcutaneous injection, oral administration, and intranasal administration without significant difference on the histopathological lesions and the tissue tropism for CEV. These findings demonstrate that ICR suckling mice are the suitable model for CEV infection, which will facilitate the investigation on CEV pathogenesis and its elicited immune response.

Histopathological examination results showed that CEV infection in mice caused severe damage to many tissues such as the myocardial interstitial edema in the heart and muscle, which is similar to the myocarditis reported for other enteroviruses [21,22,23,24]. Immunohistochemistry demonstrated a strong distribution of the CEV antigen signals in the heart and muscles, suggesting that CEV indeed targets these tissues. Additionally, strong CEV antigen signals were also detected in the lung and intestines, which is congruent with the clinical signs characterized by digestive and respiratory disorders observed in goats with CEV infections. It is worthwhile to mention that strong CEV antigen signals were revealed in the liver, which was different to the available reports on enterovirus infection [2,25]. Whether CEV-JL14 strain has a specific tissue tropism to liver, or it is a general pattern for different CEV strains is an interesting subject for future investigation.

In conclusion, we successfully established a neonatal mouse model using CEV-JL14 and demonstrated the viral pathogenicity and tropism of CEV-JL14, which provides a model system for elucidating the pathogenesis of CEV-JL14 viruses and the elicited immunity.

## Figures and Tables

**Figure 1 viruses-15-00475-f001:**
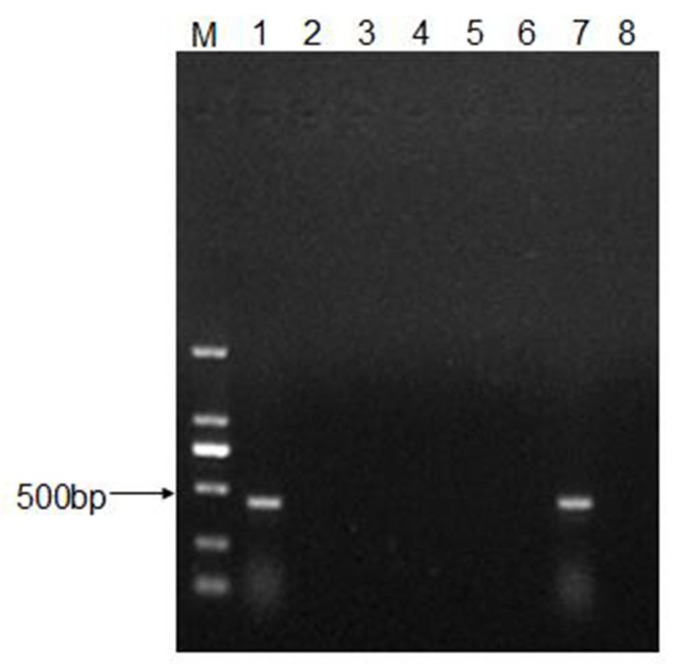
ICR suckling mice are susceptible to CEV-JL14 infection. Three-day-old ICR, BALB/c, and Kunming suckling mice were injected subcutaneously with 50 μL virus suspension containing 2 × 10^8^ TCID_50_ of CEV-JL14. The suckling mice injected subcutaneously with the same volume DMEM were used as control groups. RNAs were extracted from mixed tissue samples of mice infected by CEV-JL14 and used to amplify CEV-JL14 virus gene fragments. Fragments with expected sizes were only detected in ICR (Lane 1), while no fragments were detected in BALB/c (Lane 2) and Kunming (Lane 3) suckling mice. Lane 4–Lane 6 were the results from non-infected mice of ICR, BALB/c, and Kunming strains. Lane 7 and Lane 8 were positive and negative PCR controls, respectively. M stands for the DNA ladder.

**Figure 2 viruses-15-00475-f002:**
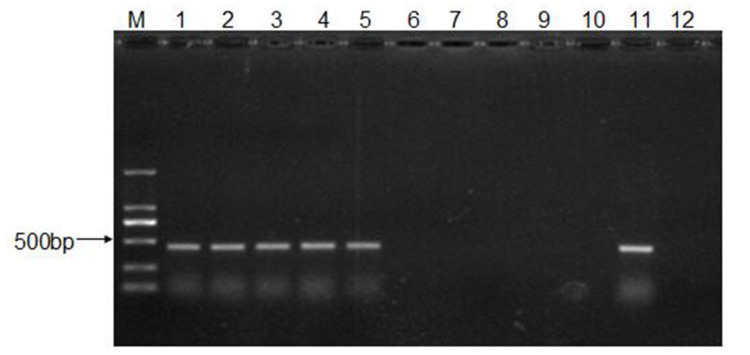
Determination of CEV-JL14 infective routes. Three-day-old ICR suckling mice in experimental groups were injected with 2 × 10^8^ TCID_50_ of CEV-JL14 virus by intraperitoneal injection, intramuscular injection, subcutaneous injection, oral administration, and intranasal administration. CEV-JL14 was detected in suckling mice inoculated with CEV-JL14 via all infective routes (Lane 1–5). No fragments were detected in control groups (Lane 6–10). Lane 1: intraperitoneal injection; Lane 2: intramuscular injection; Lane 3: subcutaneous injection; Lane 4: oral administration; Lane 5: intranasal administration. Lane 11 and Lane 12 were positive and negative control, respectively. M stands for the DNA ladder.

**Figure 3 viruses-15-00475-f003:**
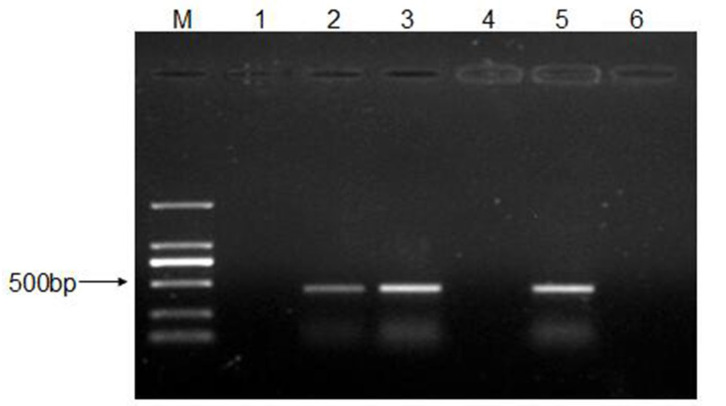
The minimal infective dose for CEV-JL14 to ICR suckling mice. Three-day-old ICR suckling mice were injected with 2 × 10^4^, 2 × 10^6^, and 2 × 10^8^ TCID_50_ CEV-JL14 by subcutaneous injection, respectively. Mixtures of tissue samples were collected at 5 dpi for RT-PCR to amplify CEV-JL14 genomic fragments. No fragment was observed in mice infected by 2 × 10^4^ TCID_50_ viruses (Lane 1). Fragments from ICR suckling mice inoculated with 2 × 10^6^ (Lane 2) and 2 × 10^8^ TCID_50_ (Lane 3) CEV-JL14 were amplified with expected size. No fragments were detected in the control groups (Lane 4). Lane 5 and Lane 6 were positive and negative control, respectively. M stands for the DNA ladder.

**Figure 4 viruses-15-00475-f004:**
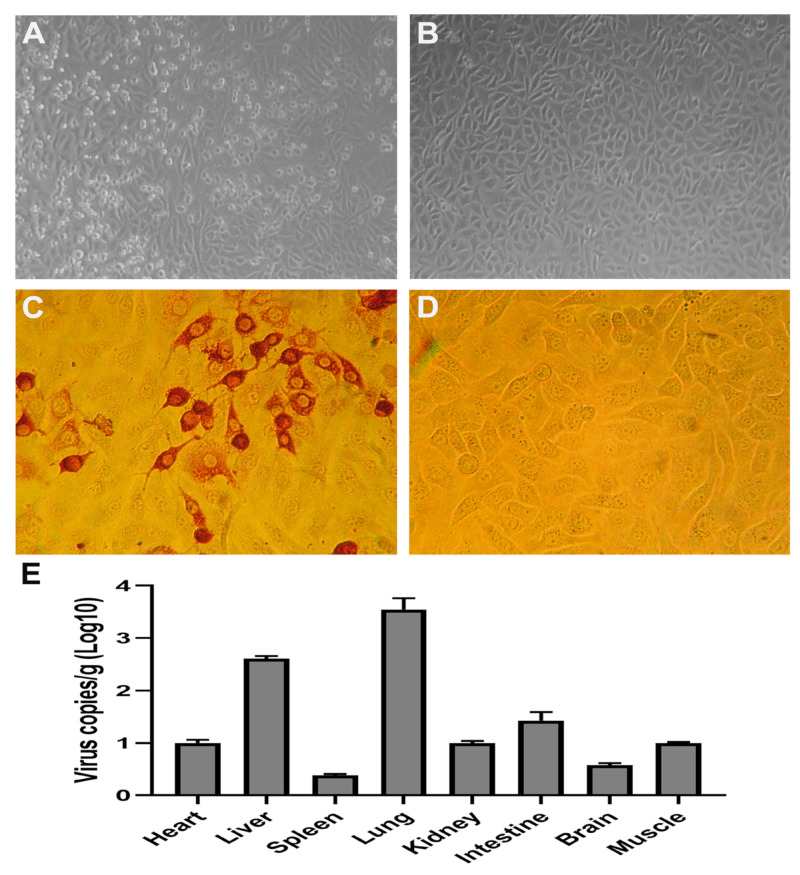
Recovery of the virus from infected mice. Tissue samples from infected mice were collected, homogenized, and used to inoculate the Vero cells. Tissue samples from non-infected mice were used as control. Typical cytopathic effects (CPE) were usually observed in the 36–48 h post inoculation (**A**). No CPEs were noticed in Vero cells inoculated with normal mice tissue samples (**B**). IPMA revealed the strong signals in Vero cells infected with CEV-JL14 (**C**). No signals were observed in the control groups (**D**). Three-day-old ICR suckling mice were injected with 2 × 10^6^ TCID_50_ CEV-JL14 by subcutaneous injection. Tissues including heart, liver, spleen, lung, kidney, intestine, brain, and muscle were harvested after 5 dpi for RT-qPCR. Viral loads in the corresponding tissues from three infected mice (5 dpi) by CEV-JL14 virus were quantitated and represented as average virus copies of log 10/g tissue (**E**).

**Figure 5 viruses-15-00475-f005:**
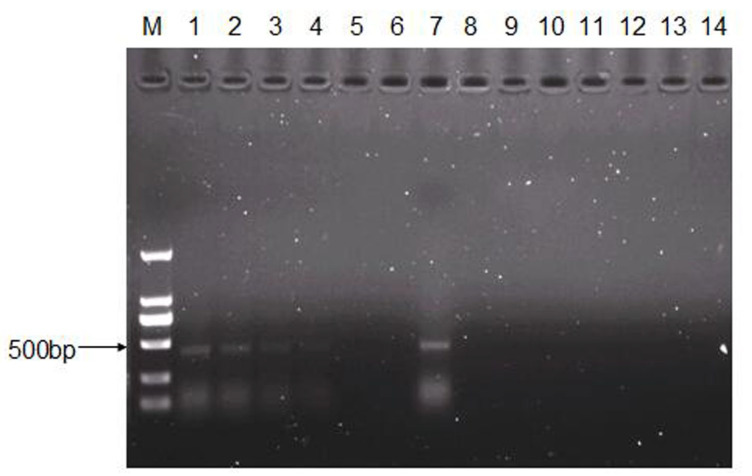
Persistence of CEV-JL14 infection revealed in ICR suckling mice. Three-day-old ICR suckling mice were injected with 2 × 10^6^ TCID_50_ CEV-JL14 by subcutaneous injection. Suckling mice in the control groups were injected with DMEM. Tissues samples were collected at 5, 11, 14, 16, 18, and 21 dpi, respectively. RT-PCR were performed to amplify CEV-JL14 genomic fragments. Fragments with expected size were present at 5 (Lane 1), 11 (Lane 2), 14 (Lane 3), and 16 dpi (Lane 4) in infected mice. No fragments were observed at 18 (Lane 5) and 21 dpi (Lane 6) in infected mice. Similarly, no fragments were seen at 5 (Lane 9), 11 (Lane 10), 14 (Lane 11), 16 (Lane 12), 18 (Lane 13), and 21 dpi (Lane 14) in control mice. Lane 7 and Lane 8 were positive and negative control, respectively. M stands for the DNA ladder.

**Figure 6 viruses-15-00475-f006:**
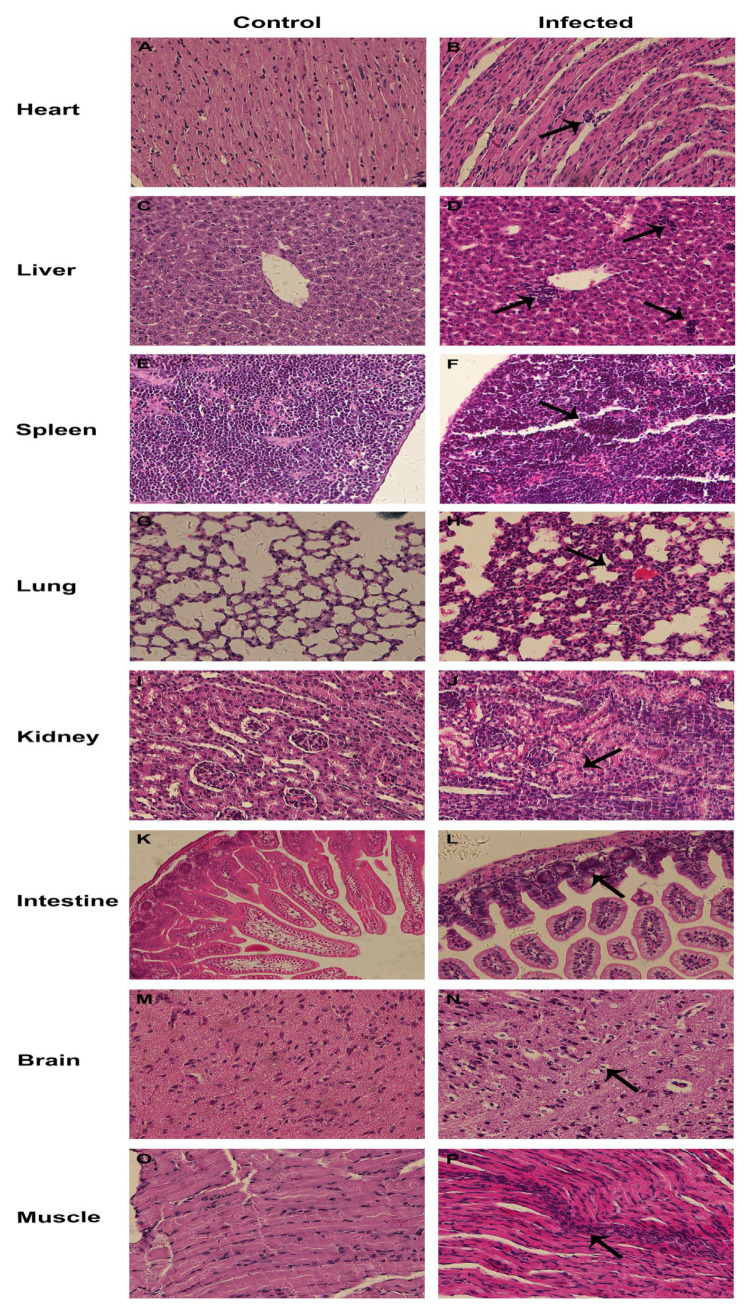
Histopathological lesions revealed in mice infected with CEV-JL14. Three-day-old ICR suckling mice were injected with 2 × 10^6^TCID_50_ CEV-JL14 via different infective routes. The mice were euthanized 5 dpi. Tissue samples were collected and processed for histopathological examination using H&E staining. The representative results are shown. Obvious histopathological lesions and inflammatory cell infiltration were observed in the heart (**B**), liver (**D**), spleen (**F**), lung (**H**), kidney (**J**), intestine (**L**), brain (**N**), and muscle (**P**), as indicated by arrow in comparison with corresponding tissues in control mice (**A**,**C**,**E**,**G**,**I**,**K**,**M**,**O**). Tissues including the heart (**B**), liver (**D**), and kidney (**J**) were obtained from mice inoculated via intramuscular injection. Spleen (**F**) and muscle (**P**) were from mice inoculated via intraperitoneal injection. Lung (**H**) was harvested from mice inoculated via intranasal administration. Intestine (**L**) was collected from mice inoculated via subcutaneous injection. Brain (**N**) was from mice inoculated via oral administration.

**Figure 7 viruses-15-00475-f007:**
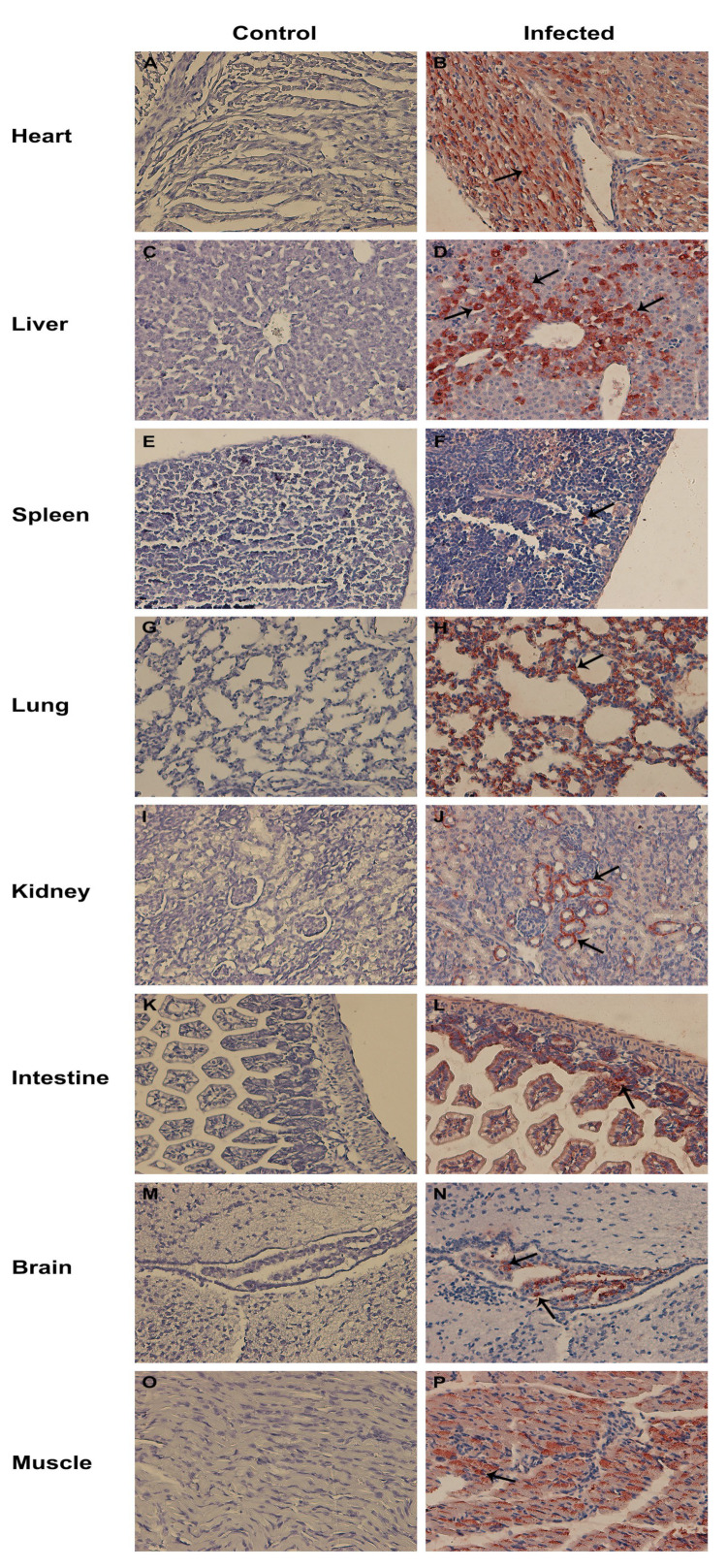
Representative results showing the viral tissue tropism in mice infected with CEV-JL14. Three-day-old ICR suckling mice were injected with 2 × 10^6^ TCID_50_ CEV-JL14 via different infective routes. Tissue samples were collected from the infected mice (5 dpi) and processed for immunohistochemistry assay to examine the tissue tropism of CEV-JL14. CEV-JL14 antigens were observed in the majority of the tissues examined, especially in the heart (**B**), liver (**D**), spleen (**F**), lung (**H**), kidney (**J**), intestine (**L**), brain (**N**), and muscle (**P**) as indicated by arrow in comparison with corresponding tissues in control mice (**A**,**C**,**E**,**G**,**I**,**K**,**M**,**O**). Heart (**B**) was obtained from mice inoculated via intramuscular injection. Liver (**D**) and brain (**N**) were from mice inoculated via intraperitoneal injection. Tissues including spleen (**F**), kidney (**J**), and intestine (**L**) were from mice inoculated via subcutaneous injection. Lung (**H**) and muscle (**P**) were from mice inoculated via intranasal administration.

**Table 1 viruses-15-00475-t001:** Severity of histopathological lesions in mice experimentally infected with CEV-JL14 with different routes.

Tissue/Inoculation	Intraperitoneally	Intramuscularly	Subcutaneously	Gavages	Intranasally
Heart	++	++	++	+	+
Liver	++	+++	++	++	++
Spleen	++	++	++	+	+
Lung	+++	+++	++	+	+++
Kidney	++	++	+	++	++
Intestine	+++	++	+	++	+
Brain	+	++	++	+++	+
Muscle	++	++	+	+	+

“+” refers to light histopathological lesions; “++” stands for moderate histopathological lesions; “+++” refers to severe histopathological lesions.

**Table 2 viruses-15-00475-t002:** CEV-JL14 antigen detected in tissues in mice infected with different routes.

Tissue/Inoculation	Intraperitoneally	Intramuscularly	Subcutaneously	Gavages	Intranasally
Heart	+	+++	++	+	+
Liver	+++	++	++	+++	++
Spleen	+	+	++	++	+
Lung	+++	++	+++	++	+++
Kidney	+	++	++	+	++
Intestine	++	+	+++	++	+
Brain	++	+	+	++	+
Muscle	+++	++	++	++	+++

“+” refers to low rate of cells detected as CEV positive; “++” stands for relative high rate of cells detected as CEV positive; “+++” refers to many positive cells detected with CEV antigen.

## Data Availability

No new data were created or analyzed in this study. Data sharing is not applicable to this article.

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
