# Peer review of "A Neonatal Murine Model for Caprine Enterovirus Infection and the Viral Tissue Tropism"

_viruses, 2023, doi:10.3390/v15020475_

Round 1

Reviewer 1 Report

In 2017, Wang et al. isolated and characterized a novel enterovirus from goats with severe diarrhea with high morbidity and mortality. The novel virus was designated as caprine enterovirus JL14 (CEV-JL14) and was proposed as a member of the new species, Enterovirus L, within the Enterovirus genus of the family Picornaviridae (https://doi.org/10.1371/journal.pone.0174600).

Enteroviral infection in goats is an emerging infectious disease and it poses a threat to modern stock-raising. Viral pathogenicity for animal enteroviruses remains still largely unknown and developing a relatively cheap and easy-to-handle animal model system is an effective means for research and investigation.

In the submitted manuscript, the authors present the establishment of a murine model for caprine enteroviruses and describe the inoculation routes, the infective dose, the tissue tropism, and the underlying pathogenesis in mice.

To my opinion, the presented research work is relevant and significant. Suckling mice are a model system that has been widely used for the studies of human enteroviruses. Of course, it would be best if viruses could be studied in a model system of the same animal species where the virus has been isolated, but this could be a rather expensive laborious, and close to impossible task. Establishing a model system in small laboratory animals deserves sharing with the scientific community. What is more, the manuscript provides insights into the tissue tropism and the underlying pathogenesis studied on the so-established animal model.

In fact, I have no major comments. But I have some minor remarks and comments. Here they follow:

Minor remarks:

1. The authors use the terms ‘goat’ and ‘caprine’ as interchangeable synonyms. In keywords, only caprine enterovirus is indicated. Wouldn’t be more appropriate to use only one of the terms throughout the text and that term exectly to be stated in the keywords?  

2. Introduction – the authors present briefly enterovirus species A to G. It is not clear why they spare the presentation of the rest enterovirus species, i.e. H to L. It is also not clearly stated to which enterovirus species CEV-JL14 belongs.

3. Introduction, line 22: EV71 is rather the name of the virus, not of the disease it causes.

4. If there is some knowledge on the pathology/histopathology of CEV-JL14 in goats, it should be briefly mentioned in the Introduction.

5. Materials and Methods, CEV-JL14 virus and infection of neonatal mice, line 72-73: was CPE observed on 48th h p.i., when cells were harvested?

6. Materials and Methods, lines 76, 79, 80: it should be 50 μL of ‘virus suspension’.

7. Results – I think that the work would benefit if the authors shared their observations on the clinical manifestation of the infection in infected mice. Were there any symptoms of a disease? Did mice show clinical symptoms on day 5 p.i. when they were euthanized? Were there mice inoculated with the higher virus dose (2x108 TCID50) that were left to survive after day 5 p.i? Was any mortality observed in the group of infected mice?

8. Results, Recovery and characterization of CEV-JL14 virus from the infected mice, line 191: on what day p.i. was CPE observed?

9. Results, Fig. 4. The main symptom of the infection in goats is severe and often lethal diarrhea. One would expect that the most severe histopathology and viral loads would be in the intestines. In the proposed mouse model, the highest viral titers expressed as virus copies are found in the lungs and the liver. The work would benefit if there were some discussion on that.

10. Results, Persistence of CEV-JL14 in ICR suckling mice, lines 212-216: was there some clinical manifestation of the infection during the studied period? Was any mortality observed?

11. Table 1. Caption, line 251: I think that it would be more precisely to head the table with “Severity of histopathological lesions …..”

12. Abbreviations should be defined when they are first used in the text – line 39 (CEV), line 106 (HRP), line 117 (H&E), and line 184 (dpi).

13. Some minor polishing of the English language and style is still needed:

Abstract, line 18 – ‘were’ should be was;tissue’ should be in plural

Introduction, line 42 – ‘province’ should be in plural

Materials and Methods, line 61 – ‘Africa’ should be ‘African’

Materials and Methods, line 64 – replace ‘infected by’ with ‘inoculated with

Materials and Methods, line 75 – I think that ‘enterovirus’ should not be used here

Materials and Methods, line 92 – ‘were’ should be ‘was

Materials and Methods, line 97 – please improve the style of the first sentence

Fig. 1, caption, line 144, 152 – please insert ‘50 μL containing 2x108…’

Fig. 1, caption, line 145 – please insert ‘the same volume’ DMEM

Fig. 2, caption, line 164 – please insert ‘the same volume’ DMEM

Discussion, line 284 – ‘carine’ should be ‘caprine

Discussion, line 298 – I would recommend refraining from the use of ‘ideal’. I would propose ‘adequate’ or ‘convenient’, or ‘suitable’

Discussion, line 303 – replace ‘by other enterovirus’ with ‘for other enteroviruses

Reviewer 2 Report

The authors have presented a neonatal mouse model for the study of the goat enterovirus, CEV-JL14. To my knowledge, this is the first mouse model for the study of members of the enterovirus G species.  The authors demonstrate that the ICR neonatal mice were susceptible to the virus at a high virus dosage (108) but not the BALB/c or Kunming suckling mice and at this high dosage, ICR neonates were infected using intraperitoneal (IP), intramuscular, subcutaneous injection, oral or intranasal administration.  Using the subcutaneous injection, neonatal ICR mice were infected with dosages at 106 or 108 but not at 104 TCID50 units.  The authors also demonstrated the presence of infectious virus (by testing for cytopathic effect and viral antigen in Vero cells) in a wide range of tissues in the infected mice with highest titers in liver and lungs. The virus infection in neonatal mice (inoculated a dosage of 106 subcutaneously) persisted out to day 16 post infection using RT-PCR. Examination of tissues from the infected mice, demonstrated lesions and viral antigen in a wide variety of tissues, with variation depending on the route of infection. Unfortunately the authors did not describe the degree of morbidity in the mice over time nor the mortality both of which should be noted as at least one study looked at mice out to day 21 post-inoculation. As the authors are establishing a mouse model for the study of this species of enterovirus, this is essential. 

Issues with the methodology used are mostly due to a lack of identification of the dosage and route of infection in most of the tissue experiments. 

Section 3.4 and Figure 4: The authors did not identify dosage used or the route of infection or the day post-inoculation nor the specific tissue type of the tissues in which viral presence and loads were identified.  This must be noted in the figure legend.

Section 3.5 and Figure 5:  What is the sensitivity of the RT-PCR used to determine persistence in the murine model? How low a copy number can be detected in this assay?  At present this assay only demonstrates that there is virus in these mice at day 16 but it may persist longer if the sensitivity of the assay is poor. 

Sections 3.6 and 3.7.  These sections describe the pathology and presence of virus in tissues of the infected mice varying by route of infection.  However, the authors do not note the route of infection of the mice that provided the tissues in Figures 6 and 7. This should be identified for each panel even if they vary from tissue to tissue. In these studies, was the dosage of virus used to infect 106 TCID50 units per mouse?  This should be noted. Clarity on this point is essential as the authors are establishing the methodology for a murine model for these viruses. 

Minor points:

Lines 28-30: The type of disease found from enterovirus infections should be referenced.  The enterovirus chapter of the current Fields Virology is frequently used for this.

Lines 93-95: Please note the nucleotide location of the primers in the GenBank entry for CEV-JL14. 

Lines 266-267:  I think this sentence should be “CEV-JL14 antigens were detected in a small number of cells in the ventricle and glial cells of the brain.”

Line 284: Carine should be Caprine.

Line 290: “Mouse model system was” should be “Mouse model systems were”.

Line 293: Should be “of different strains, and found that the ICR mouse strain”.

Lines 308-309:  “It is worthwhile...reports on enterovirus infection” should have a reference. 

Round 2

Reviewer 2 Report

I appreciate the changes the authors have made to the manuscript.  I accept that the time course study of mortality and morbidity in this model can be adddressed in later studies.  I note however that in the reply to Question 3, the authors stated that the authors "We have previously established a RT PCR method to detect CEV with a minimal detection of 1.13 × 103 copies".  If this is in Gai et al. 2018 (reference 5), I cannot find it.  Please add a note to this effect in the methods section (2.7).  
